# Prognostic implications of left ventricular strain by speckle-tracking echocardiography in population-based studies: a systematic review protocol of the published literature

Lamia Al Saikhan,[1,2] Chloe Park,[1] Rebecca Hardy,[3] Alun Hughes[1]

[1]Institute of Cardiovascular Science, School of Life and Medical Sciences, University College London, London, UK
[2]Department of Cardiac Technology, College of Applied Medial Sciences, Imam Abdulrahman Bin Faisal University, Dammam, Saudi Arabia
[3]MRC Unit for Lifelong Health and Ageing, University College London, London, UK

**Correspondence to**
Lamia Al Saikhan;
lamia.saikhan.16@ucl.ac.uk

## ABSTRACT

**Introduction** Left ventricular (LV) strain by speckle-tracking echocardiography (STE) is a comparatively new prognostic marker. Meta-analyses relating LV strain by STE to outcomes have been conducted in selected patient-based populations with established or suspected cardiovascular (CV) diseases. However, the evidence related to population-based studies of community-dwelling individuals is uncertain. The aim of this study is to provide a comprehensive systematic review and analysis of the current available literature regarding LV strain by STE as a predictor of adverse outcomes in population-based studies.

**Methods and analyses** Thesaurus and text-word searching will be used to search two online databases (MEDLINE and EMBASE) and additional sources will be identified from citation metrics and reference lists' search. Dual search results' screening, data extraction and quality assessment will be performed. Cohort studies of community/population-based samples who have had STE and followed up longitudinally for mortal and morbid events, and published in English and peer-reviewed journals will be included. Primary outcome will be all-cause mortality whereas secondary outcomes will be composite cardiac and CV end points. Risk of bias will be assessed using Newcastle-Ottawa Quality Assessment Scale of cohort studies that will be modified as appropriate. Any arising discrepancies will be discussed and resolved through consensus.

**Ethics and dissemination** Ethical approval is not required as this is a protocol for a systematic review. The findings of this study will be presented at scientific conferences and published in a peer-reviewed journal. Any amendments to the protocol will be documented and updated in the PROSPERO registry.

**PROSPERO registration number** CRD42018090302.

## Strengths and limitations of this study

► This systematic review will evaluate the evidence related to the incremental prognostic value of left ventricular (LV) strain by speckle-tracking echocardiography (STE) in relation to mortality and cardiovascular events in community-dwelling individuals or population-based samples known to be at low risk relative to selected diseased populations.

► The results of this systematic review will add to the existing evidence on the utility of STE-based LV strain as a measure of cardiac function and risk, and influence its use in longitudinal population-based studies.

► Because of the anticipated heterogeneity in results/presentation of results and/or study design across a limited number of existing studies, there may be a limited scope for meta-analysis. When meta-analysis is possible, heterogeneity in associations between studies may be difficult to explain.

therapeutic strategies in CVD.[2] Left ventricular ejection fraction (LVEF) is a simple and widely used parameter that is most commonly measured by echocardiography.[3] Reduced LVEF is known to be associated with unfavourable outcomes in populations with established CVD[4–7] but also in people without known CVD.[8 9] However, this parameter suffers from a number of inherent limitations including load dependency, low reproducibility and geometric assumptions.[3] Furthermore, its inverse association with outcomes varies considerably across the spectrum of LVEF, being greatest in moderately to severely impaired LV systolic function.[7]

LV strain imaging, tissue Doppler based and speckle-tracking based, has emerged as a powerful and sensitive tool allowing an accurate quantification of myocardial mechanics including longitudinal, circumferential and radial shortening/lengthening, and torsion.[10]

## INTRODUCTION

Cardiovascular disease (CVD) is the leading cause of mortality worldwide and accounts for 31% of deaths.[1] Assessment of left ventricular (LV) global systolic function is central in evaluating prognosis, as well as in determining

Strain, a dimensionless index, is defined as a change in length of a myocardial segment: strain $(\varepsilon) = \Delta L / L_0$, where $\Delta L$=change in length and $L_0$=original length.[3 10 11] The major limitation of tissue Doppler-derived strain (natural strain) is the angle dependency of measurements and this technique has been superseded by speckle-tracking echocardiography (STE) (Lagrangian strain).[10] As the term implies, STE is based on the analysis of myocardial features (speckles) on grayscale B-mode images that are generated by ultrasound interference patterns within the myocardium.[10–12] These speckles are tracked frame by frame during the cardiac cycle providing a sensitive and objective measure of global and regional myocardial function (ie, Lagrangian strain).[10–12] Unlike tissue Doppler imaging (TDI), STE is angle independent, can differentiate active from passive motion[13] and has been validated in an experimental setting against sonomicrometry[14] as well as in a clinical setting against cardiac magnatic resonance imaging (CMR).[15]

STE-based LV strain imaging has been shown to have an independent and additive prognostic value over conventional echocardiographic measures in a number of studies of diseased populations. These included patients with chronic[16 17] or acute[18] heart failure (HF) including both people with reduced[19 20] and preserved[21] LVEF. Further, STE-based LV strain imaging has prognostic utility after acute myocardial infarction (MI)[22 23] and in chronic ischaemic cardiomyopathy.[24] Importantly, STE-based LV strain especially global longitudinal strain (GLS) has a potential value in detecting subclinical LV systolic dysfunction (LVSD) when LVEF is within normal limits.[3] Indeed, the most common clinical setting is the prediction of cardiotoxicity in patients receiving cancer therapy.[25] GLS has also shown an association with unfavourable outcomes in asymptomatic severe aortic stenosis[26] and hypertrophic cardiomyopathy[27] with preserved LVEF. Alterations in GLS despite preserved LVEF have been demonstrated in populations with risk factors that predispose to CVD, including ageing,[28]

diabetes mellitus (DM),[29] hypertension[30] and obesity,[31] and may be the earliest marker of LVSD.

Previous systematic reviews and meta-analyses relating LV strain by STE to outcomes have been conducted in selected patient-based samples with established[32–35] or established plus suspected CVD.[36] However, no systematic review has been performed of studies which have recruited community-dwelling individuals or population-based samples, who were not selected on the basis of disease or clinical status, and are known to be at low risk compared with selected diseased populations, and the utility of STE in this setting is uncertain. This may be important since selecting samples based on disease status can introduce bias.[37] Consequently, we sought to investigate whether LV strain measured by STE is associated with risk of total and cardiovascular (CV) mortality and morbidity independent of conventional risk factors in population-based samples. We therefore proposed to carry out a systematic review and, where possible, meta-analysis of the current literature relating LV strain by STE to mortality and CV morbidity in the general population.

## METHODS

### Reporting

This systematic review will be conducted and reported in adherence with Preferred Reporting Items for Systematic Reviews and Meta-Analyses (PRISMA) statement[38] and PRISMA protocols.[39]

### Eligibility criteria

#### Inclusion criteria

The inclusion criteria which relate to the type of study design, population and measurement procedure are summarised in table 1. Studies will only be included if they assess the prospective association of LV strain with at least one of the prespecified outcomes in community-dwelling individuals who were not selected on the basis of disease or clinical status (table 1). To be eligible for inclusion, studies will be required to include a statement

**Table 1**  Eligibility criteria of the systematic review

| Type of study design | ▶ Longitudinal (cohort) studies, including placebo limbs of population-based clinical trials |
|---|---|
| Type of participants/ population | ▶ Adults (>18 years)<br>▶ General population, community/population-based samples or community-dwelling individuals not selected on the basis of disease or clinical status |
| Type of procedure | ▶ Speckle-tracking echocardiography (STE) |
| Measured parameters (exposure) | ▶ LV strain measured at rest by STE in any direction of the myocardial motion (ie, global longitudinal strain and/or circumferential strain and/or radial strain and/or transverse strain, and/or any STE-derived parameters, for example, torsion and/or twist) |
| Type of outcomes | ▶ Primary outcome: all-cause mortality<br>▶ Secondary outcomes: composite cardiovascular and cardiac end points (see text for details) |
| Year of publication | ▶ No limit applied |
| Languages | ▶ Articles published in English |
| Publication status | ▶ Published in a peer-reviewed journal |

regarding the ethical approval or adherence to an appropriate standard (such as the Declaration of Helsinki).[40]

## Outcomes

The primary outcome will be all-cause mortality. Secondary outcomes will be (1) composite CV end points, including any combination of CV mortality, coronary heart disease (CHD) events (MI, unstable angina, angina/ischaemia requiring emergent hospitalisation or revascularisation), HF hospitalisation, new-onset atrial fibrillation (AF), life-threatening arrhythmia, recorded automatic implantable cardioverter defibrillator (AICD) shocks, stroke, transit ischaemic attack or peripheral arterial disease with arterial revascularisation procedure or (2) composite cardiac end point, including any combination of CV mortality, CHD events (MI, unstable angina, angina/ischaemia requiring emergent hospitalisation or revascularisation), HF hospitalisation, new-onset AF, life-threatening arrhythmia, recorded AICD shocks. Tertiary outcomes will be any individual secondary end point included in the composite cardiac or CV end point.

### Exclusion criteria

The exclusion criteria will be as follows: (1) studies not meeting the inclusion criteria; (2) case–control studies, cross-sectional studies or randomised controlled trials which lack a placebo limb; (3) studies in unrepresentative samples of the general population (eg, clinic populations) or patients with established/known CVD (eg, patients with MI or HF); (4) LV strain measured by TDI or by another imaging modality (eg, CMR); (5) studies with end points that do not match those specified in this review (eg, LV remodelling) or (6) abstracts, reviews, conference proceedings or letters to the editor.

### Search strategy
#### Electronic searches

This literature search will be performed systematically in the following online databases: MEDLINE (1946 to present) and EMBASE Classic + EMBASE (1947 to present) via OvidSP interface. Research has suggested that these databases are likely to be sufficient to identify relevant studies.[41] A combination of thesaurus and text-word searching will be used to comprehensively extract all relevant articles. Boolean operators (AND/OR), proximity operators and truncation commands will be used when necessary to narrow the search to the relevant literature. The effectiveness of the search strategy will be tested and refined accordingly. The relevant thesaurus terms will be developed first for the MEDLINE database and will be checked and modified for EMBASE database as appropriate. A modified version of SIGN filters for observational studies will be used to capture the required studies (ie, population-based studies only). No time or language limit will be applied to the search and the final date on which the search will be carried out will be stated. Databases' search strategy is shown in online supplementary file 1.

### Additional sources

Other sources will be identified by thoroughly searching the reference lists of all relevant articles. We will also search the citation metrics of those articles using Web of Science Core Collection for additional studies.

### Patient and public involvement

Patients and the public were not involved in designing the protocol of this systematic review.

## DATA COLLECTION AND ANALYSES
### Study selection

The search results from each database will be retrieved and merged using reference manger software (ie, in a single EndNote library). Duplicates will be identified using the command 'Find Duplicates' and will be removed before screening is commenced. Titles and abstracts will then be screened for eligibility by two researchers working independently and irrelevant articles will be dismissed. Full texts of potentially eligible articles will be retrieved and double screened. A predefined eligibility form (see online supplementary file 2) will be applied to the identified studies to determine whether the inclusion criteria are met. Reasons for exclusions will be recorded. Discrepancies will be reviewed and resolved through consensus of all reviewers. The study selection process will be summarised using PRISMA flow diagram (figure 1).

### Data extraction

We will use a predefined data extraction form (see online supplementary file 2) which will be piloted on a sample of included studies to ensure that all relevant information is captured. Data will be extracted by two researchers independent of each other and any discrepancies will be discussed in order to achieve consensus. The following data will be extracted: (1) citation details including title, type and year of publication; (2) study and participant details including name, region and design of the study, sample size, demographics (age, sex ethnicity) and follow-up duration; (3) clinical variables including the presence or absence of hypertension, DM, dyslipidaemia, smoking status and known CVD; (4) exposure details including the hardware and the software used for the acquisition and analysis as well as details regarding the measured LV strain including the direction of the analysed myocardial motion, the echocardiographic images used for the analysis, the number of LV segments involved, the number of sonographers used to perform the analysis and if intraobserver or interobserver variability was reported; (5) details of outcomes (ie, all-cause mortality, composite CV and cardiac end points) including how these were ascertained; (6) statistical methods used, for example, Cox proportional hazards regression, as well as statistics related to the association of interest, for example, HR (or rate ratio), related measures of precision (ie, 95% CI or SE) and statistical significance (p-value) and (7) information related to adjustment for potential confounders.

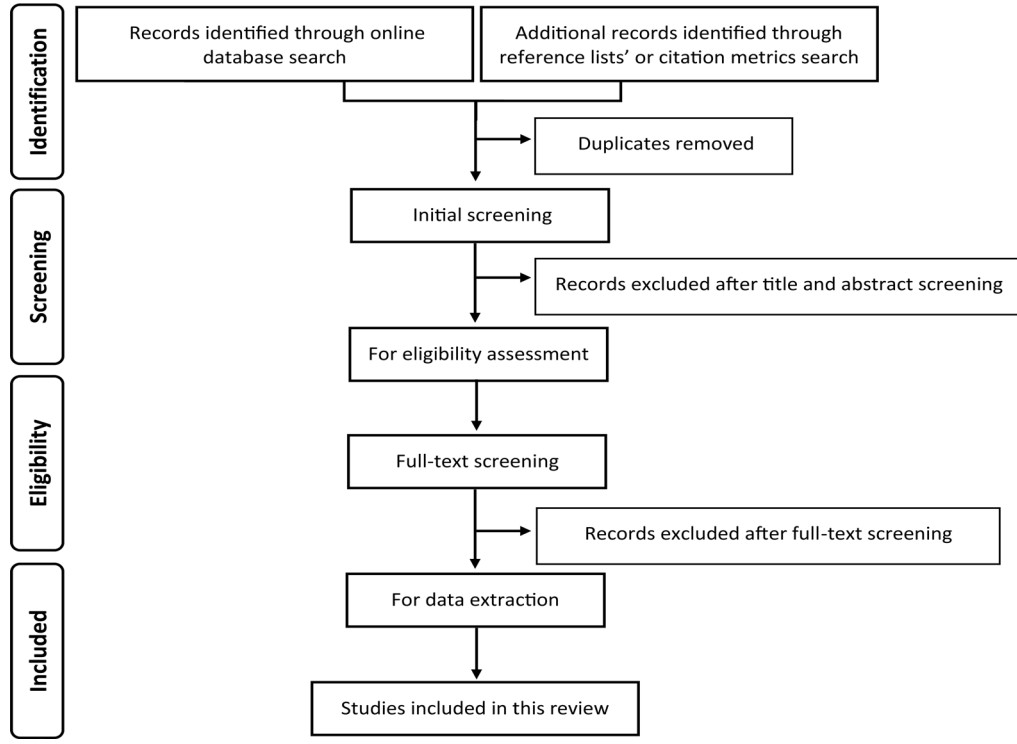

**Figure 1** Flow diagram illustrates the study selection process following the Preferred Reporting Items for Systematic Reviews and Meta-Analyses statement.[38]

## Quality assessment

Quality of included studies will be assessed using a preassigned scale (Newcastle-Ottawa Quality Assessment Scale of cohort studies)[42] that will be modified according to the review question to capture all relevant sources of potential bias (see online supplementary file 3).[43] Two reviewers will judge the quality of each study by considering the following criteria: representativeness of the cohort, completeness of collection of potential confounders, follow-up duration, assessment of outcome and the reliability of exposure (ie, LV strain). The total quality score will be reported as the average of the two researchers' scores ranging from 0, the lowest quality score, to 7, the highest quality score. Studies will be included irrespective of the quality assessment, but sensitivity analyses will be carried out to test the impact of removing the studies with the lowest quality score one at a time.

This systematic review does not study a treatment comparison, and hence quality of evidence assessment based on strict GRADE guidelines (ie, Grading Quality of Evidence and Strength of Recommendations) is not directly applicable. However, we will employ relevant components of a GRADE evaluation, namely quality of study, consistency and effect size in evaluating the strength of evidence; the issue of generalisability is dealt with through the inclusion/exclusion criteria.

## Data synthesis

Descriptive summary of the characteristics and findings of the included studies will be provided in tables. We anticipate that there may be a limited scope for meta-analysis due to differences in the method of analysis and the reporting of results across studies. If the included studies are sufficiently homogeneous and report the same statistics (eg, comparable HRs), a quantitative synthesis using random effect meta-analysis will be used to pool the results of the association of interest. Based on our expectation, we will likely to use the HR and 95% CI as an effect estimate for presenting the results and will present the data graphically in forest plots.

The degree of heterogeneity will be assessed using Higgins Thompson $I^2$ test[44 45] and Cochran's Q test.[46] If data allow, we will perform subgroup analyses or metaregression to explore prespecified potential sources of between-study heterogeneity: (1) study quality, (2) variation in exposure measurement (eg, vendors and type of strain (ie longitudinal, circumferential or radial)) and (3) variation in CV and cardiac outcomes definition. Although it may not be possible, we plan to do subgroup analyses by sex and age categories (<65 and ≥65 years). Possible publication bias will be assessed using a Funnel plot.[47]

## DISCUSSION

STE is increasingly and widely used in clinical practice. However, this technology suffers from inherent limitations, being influenced by technical as well as clinical factors.[13] High-quality images with adequate frame rates are required and there is limited evidence comparing different imaging modalities (two-dimensional and three-dimensional echocardiography). Further, this technology is vendor specific: differences between vendors

in data processing and analysis limit its generalisability and contribute to the present lack of normal values.[13 48] STE-derived strain values may also be influenced by clinical factors including age, sex and race.[13 48]

This systematic review will identify and synthesise evidence regarding STE-derived measures as prognostic indicators of mortality and CV events in population-based/community-dwelling individuals. The restriction to studies not selected on the basis of disease or clinical status should minimise potential collider bias due to index case selection. The review will also potentially highlight known limitations of STE. This analysis will add to evidence on the utility of STE as a measure of cardiac function and risk and influence its use (or otherwise) in longitudinal population-based studies and community-dwelling samples.

## ETHICS AND DISSEMINATION

This protocol is for conducting a systematic review and hence no ethical approval is required. The findings of this review will be presented at scientific conferences and published in a peer-reviewed journal.

**Contributors** LAS is the guarantor of the review who drafted the protocol and registered it in PROSPERO. LAS and AH contributed in developing the eligibility criteria, search strategy, and data extraction and quality assessment strategy. AH and CP critically reviewed and amended the protocol. RH provided statistical advice and critically reviewed and commented on the protocol.

**Funding** The authors have not declared a specific grant for this research from any funding agency in the public, commercial or not-for-profit sectors.

**Competing interests** AH works in a unit that receives support from the UK Medical Research Council (Programme Code MC_UU_12019/1) and also receives support from the British Heart Foundation (PG/15/75/31748, CS/15/6/31468, CS/13/1/30327), and the National Institute for Health Research University College London Hospitals Biomedical Research Centre. CP receives support from the British Heart Foundation (CS/15/6/31468). LA is supported by a scholarship grant from Imam Abdulrahman Bin Faisal University.

**Patient consent** Not required.

**Provenance and peer review** Not commissioned; externally peer reviewed.

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
