## [Reviewer comments · BMJ Open]

ARTICLE DETAILS

TITLE (PROVISIONAL)	Prognostic Implications of Left Ventricular Strain by Speckle-Tracking Echocardiography in Population-based Studies: A Systematic Review Protocol of the Published Literature
AUTHORS	Al Saikhan, Lamia; Park, Chloe; Hardy, Rebecca; Hughes, A

VERSION 1 – REVIEW

REVIEWER	Saraschandra Vallabhajosyula, MBBS FACP (1) Department of Cardiovascular Medicine and (2) Division of Pulmonary and Critical Care Medicine, Department of Medicine, Mayo Clinic, Rochester, Minnesota, USA
REVIEW RETURNED	14-Apr-2018

GENERAL COMMENTS	In this submission, Al Saikhan et al. present their protocol for a systematic review on existing literature evaluating left ventricular (LV) strain by speckle-tracking echocardiography in population-based studies. This is certainly of interest with the rapid advances in echocardiography technology. I have a few comments/questions for the authors' consideration: Major Comments: 1. A chief limitation of the STE technology is the difference in proprietary software between vendors resulting in inability to generalize the normal values or findings on echocardiography (please see Vallabhajosyula S et al. J Intensive Care Med. 2018 Jan 1;885066618761750. doi: 10.1177/0885066618761750. [Epub ahead of print]). The authors should seek the review the various vendors available at the current time and the strategies that they intend to use to compare these technologies. Regardless of strategies employed, this will remain a significant limitation of this systematic review.2. The authors should seek to highlight the limitation of strain imaging in their discussion - it is only possible if the 2D images are of high quality, it is still subject to biases by incomplete tracking and post-processing, the normal values remain to be validated and depend on independent echocardiographic laboratories (please see Collier P et al. J Am Coll Cardiol. 2017 Feb 28;69(8):1043-1056). Furthermore, the age/gender/race normal remain to be established. The authors should exhibit caution when analyzing sub-groups of ages, gender and race since it might be extremely heterogeneous due to the above listed reasons and the variations between individual studies. Minor Comments: 1. The authors should highlight how they seek to address the heterogeneity of data from radial, transverse and longitudinal strain.
--

	The current literature has robust evidence for global longitudinal strain (GLS) for LV function. However, the normal values for all three types remain to be established contributing further to the heterogeneity. 2. What is the rationale for the inclusion of peripheral arterial disease (PAD) events in the listed outcomes? Despite a higher propensity of PAD in patients with coronary disease, I'm not sure the events are directly related. I would recommend restricting the end-points to cardiac and cerebral vascular outcomes.
--	--

REVIEWER	Prof. N Sculthorpe University of the West of Scotland, UK
REVIEW RETURNED	25-Apr-2018

GENERAL COMMENTS	Page 7 I would prefer if the study question were more explicitly stated eg. Are STE measures independent predictors of mortality in non-clinical populations. (or a similarly structured statement). Page 9 A minor point, but the phrase 'unselected' seems peculiar (since they have been selected as participants to be part of a study). I think community dwelling, or 'apparently healthy' may be more appropriate. Page 10 Have the authors considered an upper age limit, given that GLS appears to decline in older apparently healthy adults, which may act as a confounder? For inclusion criteria, please clarify that the authors are interested in resting strain measures. Page 11 The definition of clinical populations may require some more clarification. For example, we have found (particularly in reviewing studies of older participants) that they may be classified as apparently healthy, but review of BP data indicates a substantial proportion are clearly hypertensive. This is exacerbated by the recent reclassification of hypertension thresholds. I do not think this requires substantial change, but you may need to decide if you are applying your own criteria for 'free of disease' or if you are going to rely on statements within each study. Will the authors include multi-modal studies? For example, if a study assessed STE in the general population but also provided lifestyle and diet advice? Or a nutritional supplement? Search strategy - The choice to search only 2 data bases seems somewhat limited. However the authors may have good reason for doing so, in which case the rationale should be stated. Page 13 Data extraction (4) may also need to incorporate if more than one sonographer was used, and if CoV / intra- inter variability was reported. Ethical considerations of systematic reviews can be problematic, and authors have to decide if they incorporate data collected in an
---

	unethical manner. However, I do not wish to impose onerous additional work. I suggest that the minimum standard suggested by Vergnes et al (2010) that studies should at least include a statement regarding the ethical approval, or adherence to an appropriate standard (such as the declaration of Helsinki). This is a single line addition to the data extraction form. The authors can decide whether to include or exclude studies without such a statement, but at least the 'ethical landscape' of the area can be ascertained.
--	--

VERSION 1 – AUTHOR RESPONSE

Reviewer(s)' Comments to Author:

Reviewer: 1

Reviewer Name: Saraschandra Vallabhajosyula, MBBS FACP

Institution and Country: (1) Department of Cardiovascular Medicine and (2) Division of Pulmonary and Critical Care Medicine, Department of Medicine, Mayo Clinic, Rochester, Minnesota, USA

Please state any competing interests or state 'None declared': None declared

Please leave your comments for the authors below

In this submission, Al Saikhan et al. present their protocol for a systematic review on existing literature evaluating left ventricular (LV) strain by speckle-tracking echocardiography in population-based studies. This is certainly of interest with the rapid advances in echocardiography technology. I have a few comments/questions for the authors' consideration:

Major Comments:

1. A chief limitation of the STE technology is the difference in proprietary software between vendors resulting in inability to generalize the normal values or findings on echocardiography (please see Vallabhajosyula S et al. J Intensive Care Med. 2018 Jan 1:885066618761750. doi: 10.1177/0885066618761750. [Epub ahead of print]). The authors should seek to review the various vendors available at the current time and the strategies that they intend to use to compare these technologies. Regardless of strategies employed, this will remain a significant limitation of this systematic review.

Authors:

Thank you for this comment. We agree that one of the major limitations of STE is the difference in proprietary software between vendors. Because of this limitation we appreciate that meta-analysis may not be possible and we have addressed this issue. We state that "If the included studies are sufficiently homogenous and report the same statistics (e.g. comparable hazard ratios), a quantitative synthesis using random effect meta-analysis will be used to pool the results of the association of interest". We have included differences between vendors as a potential source of heterogeneity between studies. This will be explored further by subgroup analyses or meta-regression if the data allow.

2. The authors should seek to highlight the limitation of strain imaging in their discussion - it is only possible if the 2D images are of high quality, it is still subject to biases by incomplete tracking and post-processing, the normal values remain to be validated and depend on independent echocardiographic laboratories (please see Collier P et al. J Am Coll Cardiol. 2017 Feb 28;69(8):1043-1056). Furthermore, the age/gender/race normal remain to be established. The authors should exhibit caution when analyzing sub-groups of ages, gender and race since it might be extremely heterogeneous due to the above listed reasons and the variations between individual studies.

Authors:

We have added a section in the discussion section to address this point. The aim of any subgroup analysis will be to explore any potential heterogeneity (e.g. age, gender race) in more detail. We acknowledge that subgroup analyses by age or sex will only be done "if data allow" and we will consider sources of heterogeneity between studies in making this decision.

Minor Comments:

1. The authors should highlight how they seek to address the heterogeneity of data from radial, transverse and longitudinal strain. The current literature has robust evidence for global longitudinal strain (GLS) for LV function. However, the normal values for all three types remain to be established contributing further to the heterogeneity.

Authors:

Radial (transverse), circumferential and longitudinal strain will be looked at separately, and this will be included as a potential source of heterogeneity (added in the data synthesis section).

2. What is the rationale for the inclusion of peripheral arterial disease (PAD) events in the listed outcomes? Despite a higher propensity of PAD in patients with coronary disease, I'm not sure the events are directly related. I would recommend restricting the end-points to cardiac and cerebral vascular outcomes.

Authors:

One of the aims of the study is to look at any association between STE strain and cardiovascular disease (CVD). We anticipate that most studies will report CVD as a composite outcome and that it is possible this composite will include PAD. Since we are not undertaking a participant level analysis its inclusion in the definition of CVD reflects this expectation. We wouldn't wish to exclude a study reporting CV outcomes just because PAD were incorporated in the definition.

Reviewer: 2

Reviewer Name: Prof. N Sculthorpe

Institution and Country: University of the West of Scotland, UK

Please state any competing interests or state 'None declared': None

Please leave your comments for the authors below

Page 7

I would prefer if the study question were more explicitly stated eg. Are STE measures independent predictors of mortality in non-clinical populations. (or a similarly structured statement).

Authors:

A similarly structured statement based on the reviewer's suggestion has been added.

Page 9

A minor point, but the phrase 'unselected' seems peculiar (since they have been selected as participants to be part of a study). I think community dwelling, or 'apparently healthy' may be more appropriate.

Authors:

We accept this point and have provided a clearer definition of the population (community-dwelling individuals who were not selected on the basis of disease or clinical status).

Page 10

Have the authors considered an upper age limit, given that GLS appears to decline in older apparently healthy adults, which may act as a confounder?

Authors:

We did consider it but in practice the upper age limit will probably be determined by the study and since we are not using participant level data we think we will just have to accept the effective limits imposed by the study. In addition we should make clear that we are not limiting the study to apparently healthy individual, but in population (community)-representative samples, i.e. excluding studies where the participants were selected on the basis of disease status (e.g. patients after MI or HF diagnosis). We do this to minimise index case (collider) bias – we have revised the manuscript to clarify this point.

For inclusion criteria, please clarify that the authors are interested in resting strain measures.

Authors:

Added.

Page 11

The definition of clinical populations may require some more clarification. For example, we have found (particularly in reviewing studies of older participants) that they may be classified as apparently healthy, but review of BP data indicates a substantial proportion are clearly hypertensive. This is exacerbated by the recent reclassification of hypertension thresholds. I do not think this requires substantial change, but you may need to decide if you are applying your own criteria for 'free of disease' or if you are going to rely on statements within each study.

Authors:

As we discuss above, we are not attempting to restrict to healthy (or even apparently healthy) individuals. The aim of the exclusion criteria (3) is to exclude “studies in unrepresentative samples of the general population (e.g. clinic populations) or patients with established/known CVDs (e.g. patients with MI or HF)” to minimise collider bias. As such population representative samples will include individuals with disease or subclinical disease (e.g. hypertension) We anticipate that studies may exclude those with disease at baseline and or adjust for CV risk factors in analyses and will be documented. For this categorization we will rely on statements provided by each study.

Will the authors include multi-modal studies? For example, if a study assessed STE in the general population but also provided lifestyle and diet advice? Or a nutritional supplement?

Authors:

If they meet inclusion criteria then yes, although only the placebo (not the intervention limb) of such a study would be included in any data synthesis

Search strategy - The choice to search only 2 data bases seems somewhat limited. However the authors may have good reason for doing so, in which case the rationale should be stated.

Authors:

Research has suggested that these databases are likely to be sufficient to identify relevant studies (<https://doi.org/10.1186/s12874-016-0232-1>). We will also search the citation metrics of the identified key papers using Web of Science as stated in additional sources section in addition to searching their reference lists.

Page 13

Data extraction (4) may also need to incorporate if more than one sonographer was used, and if CoV / intra- inter variability was reported.

Authors:

We have added this to the data extraction section point number 4 and data extraction form (see supplement 2).

Ethical considerations of systematic reviews can be problematic, and authors have to decide if they incorporate data collected in an unethical manner. However, I do not wish to impose onerous additional work. I suggest that the minimum standard suggested by Vergnes et al (2010) that studies should at least include a statement regarding the ethical approval, or adherence to an appropriate standard (such as the declaration of Helsinki). This is a single line addition to the data extraction form. The authors can decide whether to include or exclude studies without such a statement, but at least the 'ethical landscape' of the area can be ascertained.

Authors:

Thank you for drawing our attention to this important point. We have added this to the inclusion criteria (citing Vergnes et al.) and data extraction form (supplement 2).

VERSION 2 – REVIEW

REVIEWER	Saraschandra Vallabhajosyula, MBBS FACP (1) Department of Cardiovascular Medicine and, (2) Division of Pulmonary and Critical, Care Medicine, Department of Medicine, Mayo Clinic, Rochester, Minnesota, USA
REVIEW RETURNED	25-May-2018
GENERAL COMMENTS	No further comments.
REVIEWER	Professor N Sculthorpe Institute of Clinical Exercise and Health Science, University of the West of Scotland, Hamilton, ML3 0JB, Scotland
REVIEW RETURNED	29-May-2018
GENERAL COMMENTS	I am happy the authors have adequately addressed my concerns regarding the earlier suggested changes. No further modification required.